# GPCR-induced calcium transients trigger nuclear actin assembly for chromatin dynamics

Ying Wang[1]*, Alice Sherrard[2], Bing Zhao[3], Michael Melak[1], Jonathan Trautwein[1], Eva-Maria Kleinschnitz[1], Nikolaos Tsopoulidis [4], Oliver T. Fackler [4], Carsten Schwan[3] & Robert Grosse [3,5]*

Although the properties of the actin cytoskeleton in the cytoplasm are well characterized, the regulation and function of nuclear actin filaments are only recently emerging. We previously demonstrated serum-induced, transient assembly of filamentous actin within somatic cell nuclei. However, the extracellular cues, cell surface receptors as well as underlying signaling mechanisms have been unclear. Here we demonstrate that physiological ligands for G protein-coupled receptors (GPCRs) promote nuclear F-actin assembly via heterotrimeric $G\alpha_q$ proteins. Signal-induced nuclear actin responses require calcium release from the endo-plasmic reticulum (ER) targeting the ER-associated formin INF2 at the inner nuclear membrane (INM). Notably, calcium signaling promotes the polymerization of linear actin filaments emanating from the INM towards the nuclear interior. We show that GPCR and calcium elevations trigger nuclear actin-dependent alterations in chromatin organization, uncovering a general cellular mechanism by which physiological ligands and calcium promote nuclear F-actin assembly for rapid responses towards chromatin dynamics.

[1] Institute of Pharmacology, BPC Marburg, University of Marburg, Karl-von-Frisch-Straße 2, 35043 Marburg, Germany. [2] School of Cellular and Molecular Medicine, Biomedical Sciences, University of Bristol, University Walk, Bristol BS8 1TD, UK. [3] Institute of Experimental and Clinical Pharmacology and Toxicology, University of Freiburg, Albertstraße 25, 79104 Freiburg, Germany. [4] Department of Infectious Diseases, Integrative Virology, CIID, Heidelberg University Hospital, Im Neuenheimer Feld 344, 69120 Heidelberg, Germany. [5] CIBSS Centre for Integrative Biological Signaling Studies, University of Freiburg, Freiburg, Germany. *email: wangyin4@staff.uni-marburg.de; robert.grosse@pharmakol.uni-freiburg.de

The dynamic reorganization of actin filaments in the cytoplasm plays fundamental and well-established roles in mechanical support, membrane dynamics, and intracellular trafficking[1]. In contrast, its roles and functions in the nuclear compartment of somatic cells are only beginning to emerge[2]. Recently, filamentous actin structures were discovered in the nucleus when cells undergo serum stimulation, T-cell activation, DNA damage, or mitotic cell division[3–5]. The function of dynamic nuclear F-actin assembly in these processes ranges from regulation of gene transcription, homology-directed DNA repair to nuclear volume expansion, and chromatin decondensation[6–10]. Yet how cells transduce the stimulation by extracellular signals from the plasma membrane to the nucleus for the dynamic polymerization of nuclear actin filaments remains largely elusive.

Heterotrimeric G proteins are key components of transmembrane signal transduction via G-protein-coupled receptors (GPCRs), controlling a plethora of vital physiological functions. The different Gα subunits of heterotrimeric G proteins define specific agonist-induced GPCR signaling events. For example, $G\alpha_{12/13}$ activates downstream effectors that promote RhoA activity, while $G\alpha_{q/11}$ activation leads to the release of calcium from intracellular stores[11]. Calcium is a central second messenger of signal transduction in virtually all cells. Intracellular calcium influences essential cellular processes[12] often involving calcium-sensing proteins such as calmodulin[13]. Of note, calcium transients can be released locally in the perinuclear region from the endoplasmic reticulum (ER) and transmitted into the nucleus[14].

The actin regulator inverted formin 2 (INF2) is emerging as a factor controlled by calcium signaling that promotes perinuclear F-actin in response to increased intracellular calcium levels[15,16]. Here, we identify an important nuclear role for INF2 in response to GPCR-calcium signaling elicited by physiological ligands. Our study reveals that several GPCR agonists, such as thrombin, LPA, or ATP trigger transient bursts of nuclear F-actin formation downstream of $G\alpha_{q/11}$ to promote rapid changes in chromatin dynamics.

## Results

### GPCR ligands and calcium transients induce NAA.
We previously described serum-induced nuclear actin assembly, hereafter referred to as NAA, that occurred within 15–20 s after serum stimulation[6]. Given the rapid and transient kinetics of serum-induced NAA, we tested whether induction of calcium elevations triggers nuclear actin dynamics. NIH3T3 cells stably expressing nAC-GFP, a genetically encoded anti-actin nanobody that is derived from the actin-chromobody fused to a nuclear localization sequence[9], were stimulated with the calcium ionophore A23187 to increase intracellular calcium levels. Interestingly, addition of A23187 triggered the dynamic assembly of nuclear F-actin in 73.6% ± 7.6% (s.e.m.) cells (Fig. 1a; Supplementary Movies 1, 2). These nuclear F-actin structures appeared rapidly and were short-lived (between 60 and 120 s), similar to previously reported serum-induced NAA[6], which are different from the longer-lasting nuclear actin filaments observed during mitotic exit[7]. As LPA is abundant in serum[17], we tested the small molecule inhibitor Ki16425 to block GPCRs of the LPA receptor family 1, 2, and 3[18]. This resulted in the inhibition of serum-induced NAA (Fig. 1b). LPA receptor inhibition did not interfere with A23187-induced NAA, while the calcium chelator BAPTA-AM blocked serum-induced NAA (Fig. 1c), indicating that the induction of nuclear F-actin formation is mediated by GPCRs and calcium. We next tested for the ability of additional physiological GPCR ligands known to promote calcium elevations[11]. NAA was potently stimulated by the GPCR agonists LPA,

thrombin or ATP (Fig. 1d, e; Supplementary Movies 3, 4), indicating an instrumental role of GPCR signaling for immediate actin dynamics in the nuclear compartment.

### NAA depends on $G\alpha_{q/11}$-mediated calcium transients.
Since G proteins of the $G\alpha_{q/11}$ family mediate $Ca^{2+}$ release from the ER[11], we investigated if $G\alpha_{q/11}$ is responsible for ligand-induced NAA. Silencing of $G\alpha_{q/11}$ but not $G\alpha_{12/13}$ proteins significantly interfered with NAA elicited by the GPCR ligands LPA or thrombin (Fig. 2a, b). In order to test whether store-derived calcium elevations mediate NAA, we used the drug thapsigargin to block the ability of cells to pump calcium into the ER[19]. Interestingly, thapsigargin readily triggered NAA, while inhibition of LPA receptors using Ki16425 did not inhibit NAA induced by thapsigargin (Fig. 2c), demonstrating a direct role for intracellular calcium stores.

To address whether calcium transients precede NAA, we monitored intracellular as well as nuclear calcium elevations using the fluorescent calcium sensor GCaMP6f[20] in cells stably expressing nAC-mCherry (Fig. 2d). Addition of thapsigargin led to a transient increase of intracellular calcium levels as reflected by the cytoplasmic and nuclear increase of GCaMP6f fluorescence intensity (Fig. 2d; Supplementary Movie 5). This elevation of calcium was accompanied by the subsequent appearance of nuclear actin filaments (Fig. 2d; Supplementary Movie 5). To analyze the relationship between calcium elevation and NAA in a temporal manner, we measured the increase of fluorescence intensity of GCaMP6f and the signal heterogeneity of nAC-mCherry reflecting F-actin formation in the nucleus. An initial simulation was performed to validate the approach (Supplementary Movie 6). Subsequent data analysis showed that the appearance of NAA followed calcium elevations in the cytoplasm and nucleus after induction by thapsigargin or A23187 (Fig. 2e). Collectively, these data demonstrate that $G\alpha_{q/11}$/calcium signaling promotes NAA in intact living cells.

### INF2 is required for $Ca^{2+}$-induced NAA.
INF2 was previously linked to calcium signaling[15,16]. Western blot analysis detected endogenous INF2 in nuclear fractions from NIH3T3 cells (Supplementary Fig. 1 A). Subcellular localization of INF2 was further analyzed by expressing GFP-INF2-CAAX and a Lamin-Chromobody-mCherry[7] to detect Lamin A/C that resides below the INM[21]. This revealed a colocalization of INF2 and Lamin A/C (Fig. 3a), implicating a role for INF2 at the INM. Localization of INF2 to the nuclear membrane was confirmed by visualizing endogenous INF2 (Supplementary Fig. 1B, C). Consistent with this, we observed the formation of linear actin filaments elongating from the INM (Fig. 3b; Supplementary Movie 7). Polymerization started ~16 s after addition of A23187 and persisted for ~50 s before filaments depolymerized within 1 min, indicating that the INM was the predominant structure for initiating NAA. This further suggests that the INM may serve as a signaling hub for the initiation of nuclear actin polymerization in mammalian cells.

We next investigated whether calcium-induced NAA requires INF2. For this, INF2 was depleted by siRNA or deleted using CRISPR/Cas9 genome editing (Supplementary Fig. 1 C, D). This resulted in robust and significant inhibition of NAA upon stimulation with A23187, LPA or thrombin (Fig. 3c–g; Supplementary Movies 8–20, Supplementary Fig. 2). The crucial importance of INF2 was confirmed by rescue experiments of INF2-depleted cells while reintroducing BFP-tagged INF2 before treatment of either A23187, LPA or thrombin (Fig. 3e–g; Supplementary Movies 12–20, Supplementary Fig. 2).

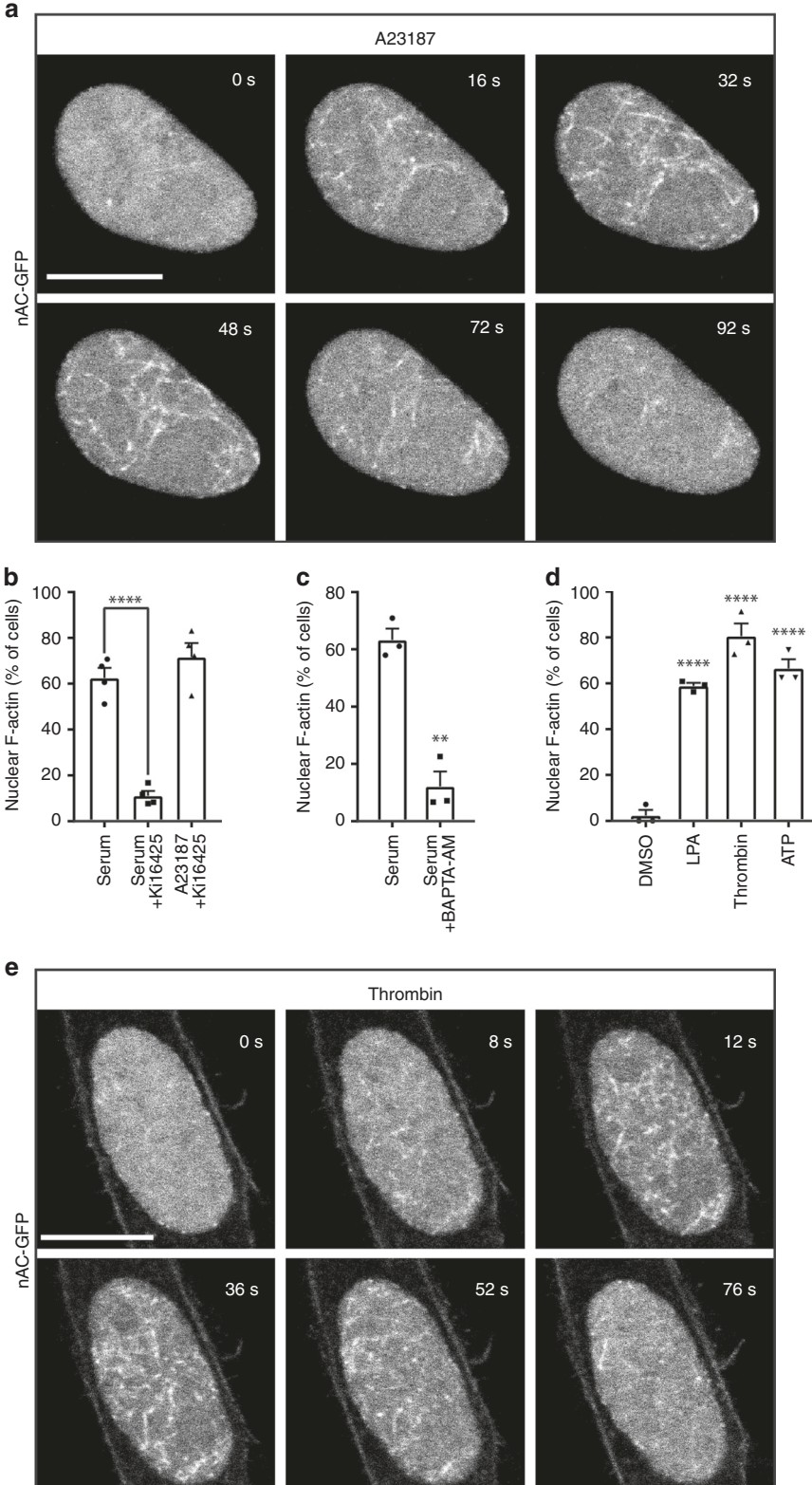

**Fig. 1** Calcium transients induce nuclear F-actin assembly. **a** Representative images of NIH3T3 cells stably expressing nAC-GFP after A23187 stimulation. In total, 73.6% ± 7.6% (s.e.m.) cells exhibited NAA. Each experiment included three samples in parallel, and about 30 cells were recorded and counted each time. Experiments were repeated three times ($n = 3$). Scale bar: 10 µm. **b** NIH3T3 cells stably expressing nAC-GFP were stimulated with 20% serum or A23187 (750 nM) after pre-treatment with or without LPA receptor inhibitor Ki16425 (20 µM) at the confocal microscope. $n = 4$ independent experiments. One-way ANOVA test, ****$p \leq 0.0001$. **c** NIH3T3 cells stably expressing nAC-GFP were stimulated with 20% serum after pre-treatment with or without BAPTA-AM (10 µM). $n = 3$ independent experiments. Two-sided Student's $t$ test, **$p \leq 0.01$. **d** NIH3T3 cells stably expressing nAC-GFP were stimulated with LPA (20 µM), thrombin (0.2 U/mL), or ATP (10 µM). $n = 3$ independent experiments. One-way ANOVA test, ****$p \leq 0.0001$. Error bars: +s.e.m. **e** Representative images of NIH3T3 cells stably expressing nAC-GFP after thrombin stimulation. Experiments were performed three times, and about 30 cells were recorded each time. Eighty percent of the cells were positive for NAA as shown in panel **d**. Scale bar: 10 µm. Source data are provided as a Source Data file

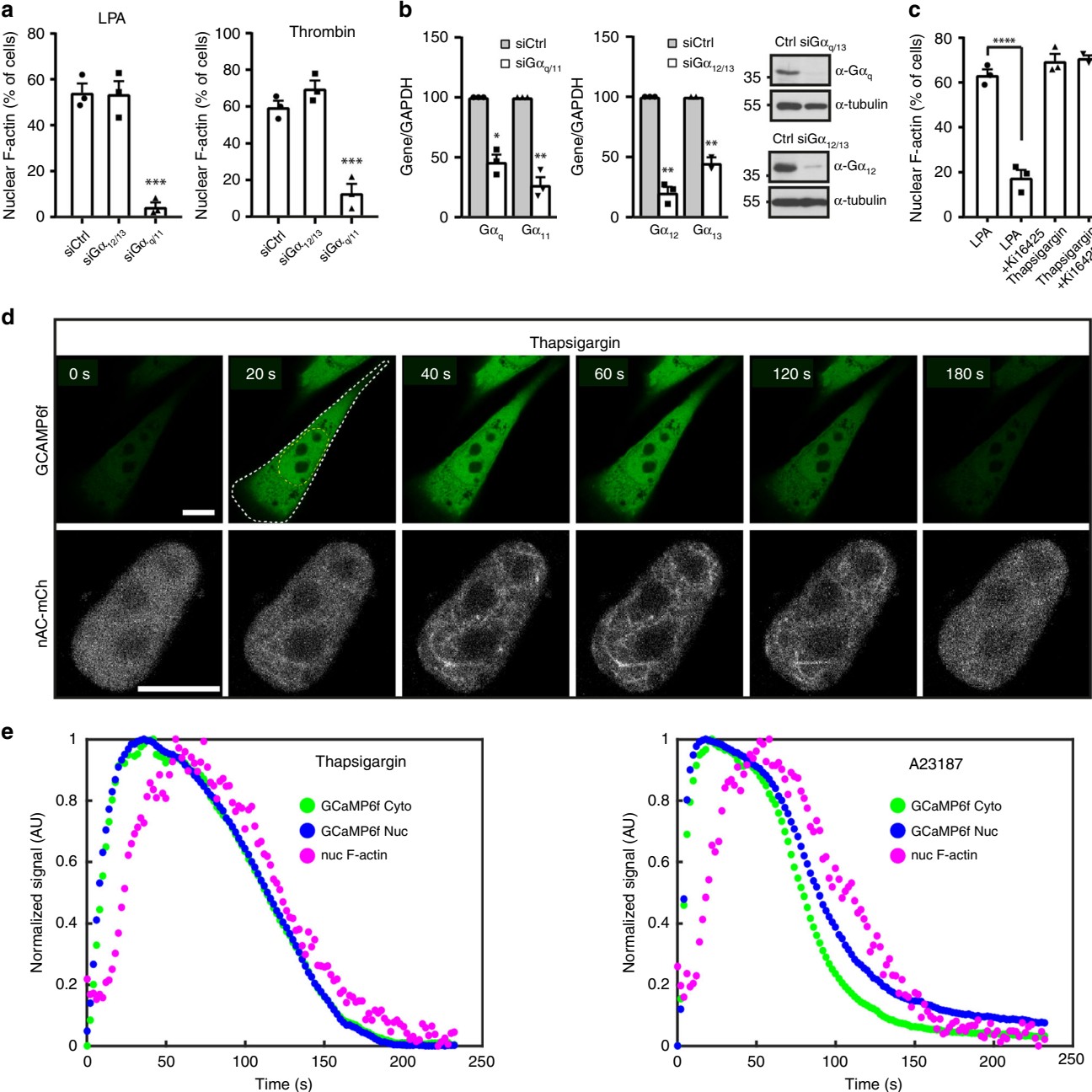

**Fig. 2** GPCR ligands induce nuclear F-actin via $G\alpha_{q/11}$ and calcium. **a** After transfection with different siRNAs, NIH3T3 cells stably expressing nAC-GFP were stimulated with LPA or thrombin. $n = 3$ independent experiments. One-way ANOVA test, ***$p \leq 0.001$. **b** Western blots and qPCR showing knockdown efficiency of respective siRNAs. Two-sided Student's $t$ test, *$p \leq 0.05$, **$p \leq 0.01$. **c** NIH3T3 cells stably expressing nAC-GFP were stimulated with LPA or thapsigargin after pre-treatment with or without Ki16425. $n = 3$ independent experiments. One-way ANOVA test, ****$p \leq 0.0001$. Error bars: +s.e.m. **d** After transfection with GCaMP6f, NIH3T3 cells stably expressing nAC-mCherry were stimulated with thapsigargin. Image sequences from a representative cell out of 35 NAA-positive cells are shown. White and yellow dashed lines indicate the region of interest for quantification. Scale bars: 10 μm. **e** Quantification of the fluorescent signals after thapsigargin or A23187 treatment showing nuclear F-actin formation after intracellular calcium release. Source data are provided as a Source Data file

**A role for $Ca^{2+}$/calmodulin and INF2 in the nucleus.** Calcium transients relay downstream signaling events through calcium sensor proteins such as calmodulin (CaM)[12,13]. By analyzing the primary sequence of INF2 we identified a putative binding site located within diaphanous-inhibitory-domain (DID) of INF2 (INF2-DID). INF2 as well as INF2-DID could be immunoprecipitated with CaM from cell extracts (Fig. 4a–c). Notably, this interaction was disrupted by CaM, in which the four calcium-binding sites D20A, D56A, D93A, D129A were mutated

(Fig. 4b)[22], evidencing that INF2 is regulated by calcium signaling through interactions with $Ca^{2+}$/CaM.

CaM resides both in the cytoplasm and nucleus[23]. To further explore whether nuclear $Ca^{2+}$/CaM plays a role in ligand-induced NAA, we expressed the nuclear CaM inhibitory protein CaMBP4-mCherry[24]. CaMBP4 expression significantly reduced NAA elicited by intracellular calcium elevations using A23187 or thrombin (Fig. 4d) arguing for an involvement of a nuclear $Ca^{2+}$/CaM-INF2 function.

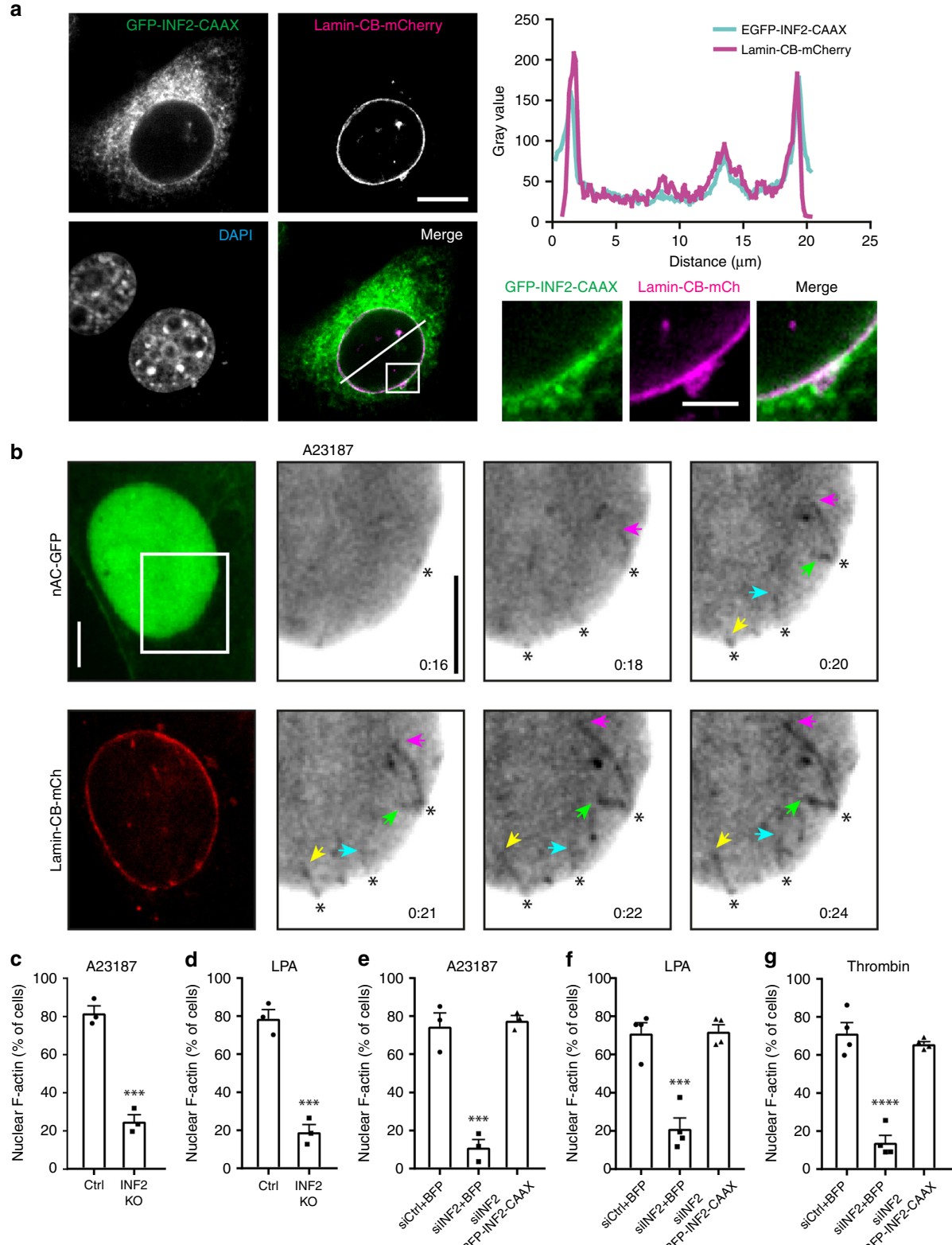

**Fig. 3** Calcium-induced nuclear F-actin depends on INF2. **a** NIH3T3 cells were transfected with GFP-INF2-CAAX and Lamin-Chromobody (CB)-mCherry and fixed (scale bar: 10 μm). Images from one representative cell out of 18 cells are enlarged (scale bar: 3 μm), and gray value plot was drawn showing colocalization of INF2 and Lamin A/C on the nuclear envelope. **b** NIH3T3 cells stably expressing nAC-GFP were transfected with Lamin-CB-mCherry and stimulated by A23187 to visualize nuclear F-actin polymerization. Arrowheads show the tips of actin filaments originating from nuclear envelope. Scale bar: 5 μm. **c**, **d** INF2 was knocked out (INF2 KO) by CRISPR/Cas9 genome editing in NIH3T3 cells stably expressing nAC-GFP. Both control and knockout cells were stimulated with A23187 or LPA. $n = 3$ independent experiments. Two-sided Student's $t$ test, ***$p \leq 0.001$. **e**–**g** INF2-silenced cells were rescued by BFP-INF2-CAAX isoform after A23187 ($n = 3$), LPA ($n = 4$) or thrombin ($n = 4$) treatment. Both control and INF2 siRNAs are modified on the sense strand with 3′-AlexaFluor 647. INF2 siRNA targets the 3′-UTR. One-way ANOVA test, ***$p \leq 0.001$, ****$p \leq 0.0001$. Error bars: +s.e.m. Source data are provided as a Source Data file

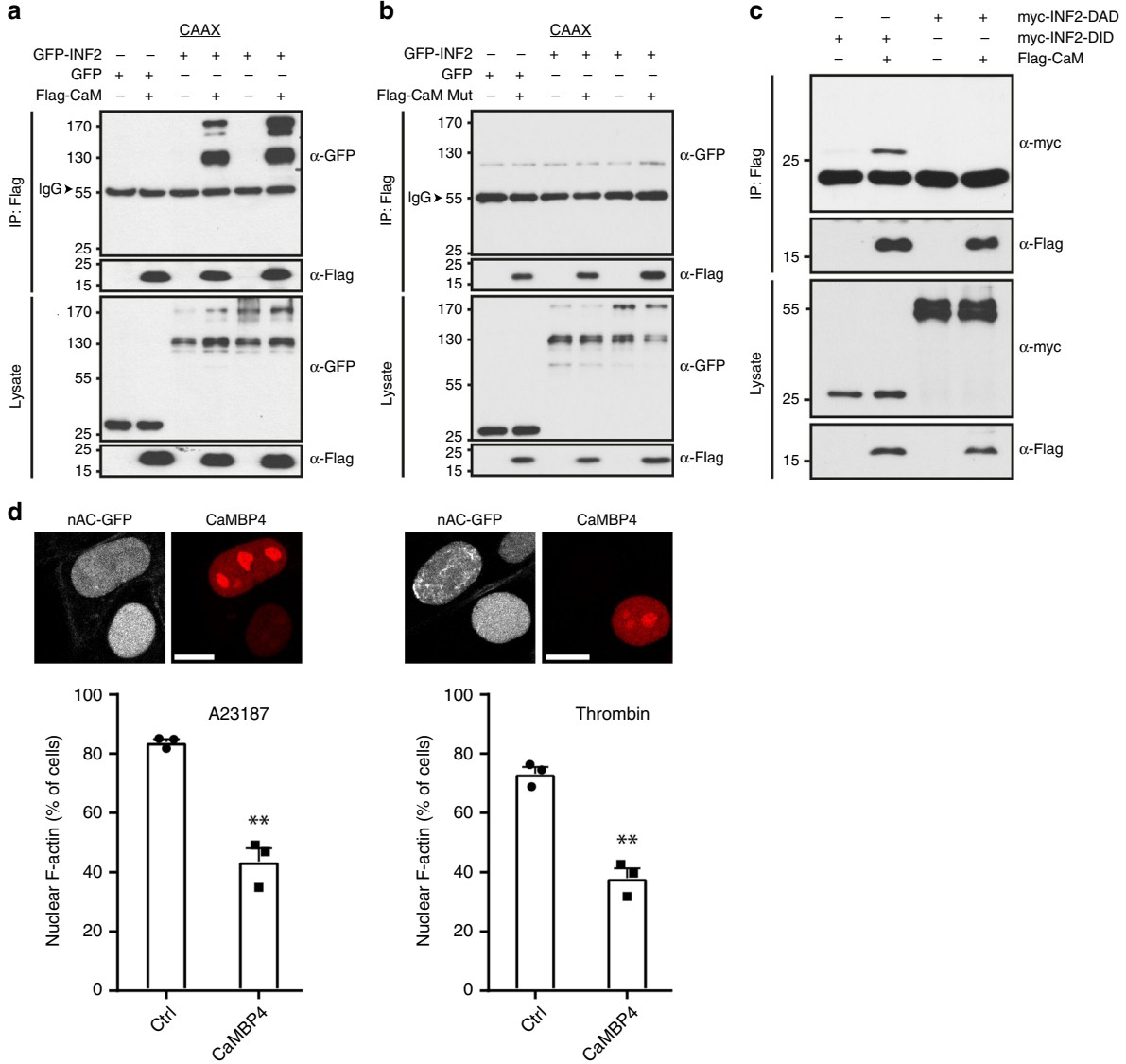

**Fig. 4** INF2 interacts with calmodulin and Ca$^{2+}$/CaM-INF2 plays a role in NAA. **a** Immunoprecipitations of HEK293 cells expressing Flag-calmodulin (CaM) and GFP-INF2. Associated INF2 was detected using anti-GFP antibodies. **b** Immunoprecipitations of HEK293 cells expressing Flag-CaM-mut (calcium-binding mutant) and GFP-INF2. Associated INF2 was detected using anti-GFP antibodies. **c** Immunoprecipitations of HEK293 cells expressing Flag-CaM and myc-INF2-DID or DAD (Diaphanous Autoregulatory Domain). Associated INF2 was detected using anti-myc antibodies. **d** Cells stably expressing nAC-GFP were transfected with mCherry or nuclear CaM inhibitor mCherry-CaMBP4. Cells were stimulated with A23187 or thrombin and counted for the formation of nuclear F-actin. $n = 3$ independent experiments. Two-sided Student's $t$ test, **$p \leq 0.01$. Error bar: +s.e.m. Representative images are shown above. Scale bars: 10 μm. Source data are provided as a Source Data file

**GPCR/Ca$^{2+}$ induced chromatin dynamics via INF2 and NAA.** We previously showed that nuclear F-actin promotes chromatin reorganization after mitosis[7]. To explore the functional impact of GPCR/calcium toward chromatin dynamics, we employed a fluorescence lifetime imaging microscopy (FLIM) assay to measure chromatin compaction in intact cells by determining GFP fluorescence lifetime between GFP- and mCherry-histones[7,25]. Interestingly, chromatin became significantly more dynamic upon treatment of A23187 or thrombin (Fig. 5a, b), while no alterations could be observed in INF2 knockout cells (Fig. 5c, d). Similar results were obtained using siRNA against INF2 (Supplementary Fig. 3A, B). In addition, electron microscopy of cryopreserved cells demonstrated electron-dense structures in control cells, while electron-dense structures were reduced when cells were stimulated by A23187 or thrombin (Fig. 5e; Supplementary Fig. 3 C), indicating a more dynamic nature of

chromatin in response to GPCR agonists or calcium stimuli, which could not be observed in INF2 knockout cells (Fig. 5f; Supplementary Fig. 3C). To investigate a requirement for nuclear actin polymerization, we expressed a previously validated nuclear, non-polymerizing actin mutant R62D (NLS-Flag-actin-R62D)[7]. Indeed, expression of NLS-R62D significantly inhibited chromatin dynamics in response to A23187 or thrombin stimulations (Fig. 5g, h). These data demonstrate a requirement of INF2 and polymeric nuclear actin for GPCR/calcium-triggered chromatin organization.

**INF2 and mDia cooperate in NAA.** INF2 is an important disease-associated formin that has been causatively linked to kidney diseases and neuropathies[26]. INF2 was shown to physically interact with formins of the mDia family[27]. We have

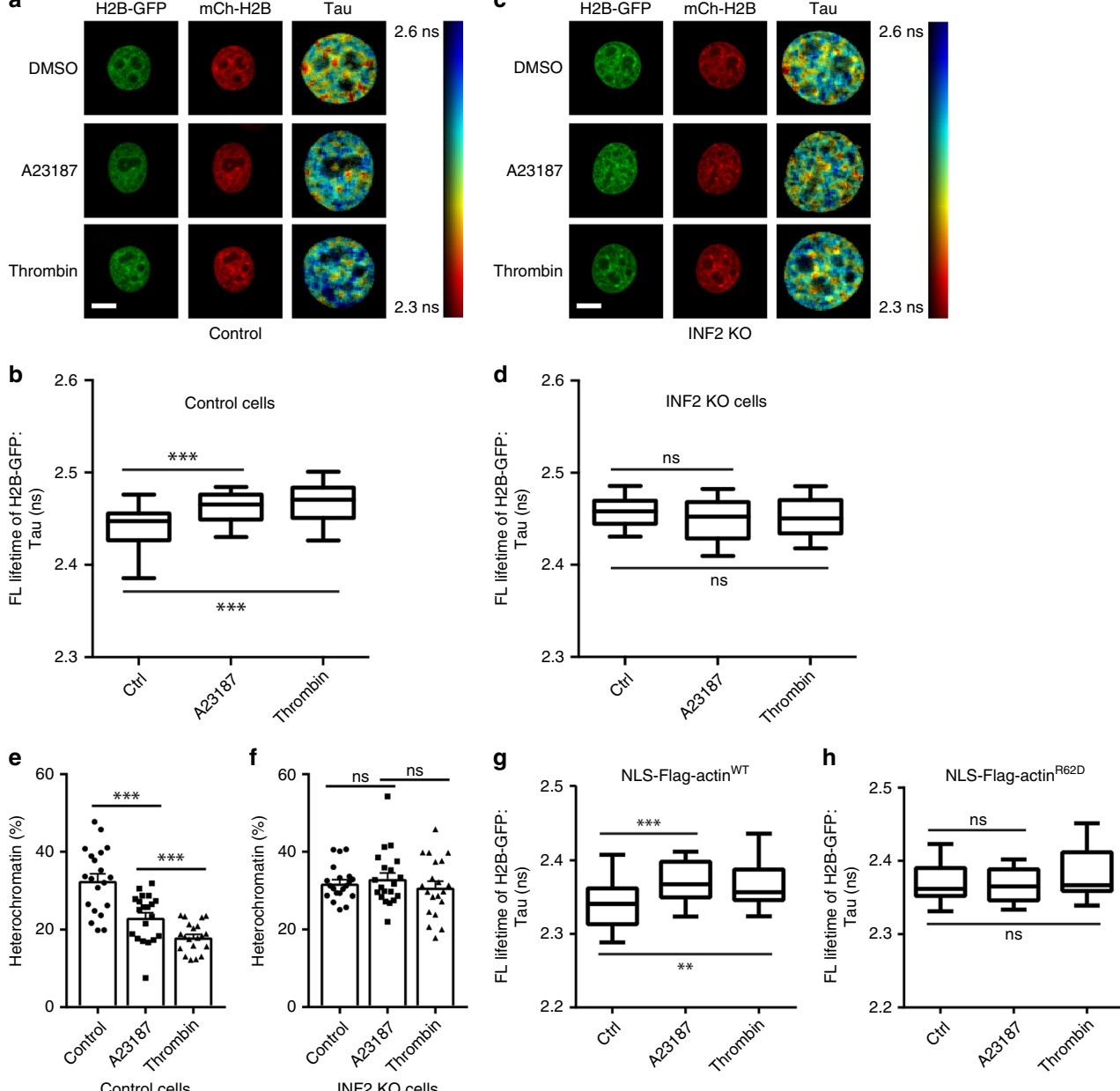

**Fig. 5** INF2 and nuclear F-actin are required for chromatin dynamics. **a**, **c** Representative images and fluorescence lifetime maps of control and INF2 KO cells expressing H2B-GFP and mCherry-H2B, where the ruler on the right depict GFP fluorescence lifetime. Cells were treated with DMSO, A23817, or thrombin for 15 min, before being fixed. Scale bars: 10 μm. **b**, **d** Control and INF2 KO cells expressing H2B-GFP and mCherry-H2B were measured for GFP fluorescence lifetime. Cells were treated with DMSO, A23187, or thrombin for 15 min. Thirty cells were measured for each condition. Boxes extend from the 25th to 75th percentiles. Middle line indicates median. Whiskers represent min to max with all points shown. One-way ANOVA test, ***$p \leq 0.001$. **e**, **f** Cryopreserved cells were quantified for condensed chromatin based on EM images. $n = 20$ cells. One-way ANOVA test, ***$p \leq 0.001$. Error bars: +s.e. m. **g**, **h** FLIM analyses of Flag-NLS-actin WT and Flag-NLS-actin-R62D cell nucleus showing chromatin reorganization after A23187 or thrombin. $n = 30$ cells for each condition. Boxes extend from the 25th to 75th percentiles. Middle line indicates median. Whiskers represent min to max with all points shown. One-way ANOVA test, **$p \leq 0.01$, ***$p \leq 0.001$. Source data are provided as a Source Data file

previously demonstrated that mDia formins are involved in serum-induced NAA[6]. Consistent with this, interfering with mDia and INF2 resulted in additive inhibitory effects towards calcium-induced NAA (Supplementary Fig. 3D–F), suggesting a cooperative function of both formins.

## Discussion

GPCRs are major clinical drug targets in a plethora of pharmacological treatments and are activated by extracellular signals such as hormones, neurotransmitters, or sensory stimuli. As such, G-protein-mediated second messenger responses play a central role in cell and tissue homeostasis. Our data indicate that GPCR signaling results in rapid and transient actin filament assembly in the nuclear compartment to promote changes in chromatin dynamics. Thus, one downstream consequence of this pivotal transmembrane signaling pathway is to regulate ligand-elicited nuclear actin reorganization responses.

In addition to the role of monomeric actin in chromatin remodeling complexes, recent findings support a role of nuclear

actin polymerization in regulating spatiotemporal chromatin organization and the maintenance of nuclear architecture[7] or the movement of double-stranded break clusters for homology-directed repair[8,10]. Our findings reveal a general mechanism by which cells readily communicate information from physiological GPCR ligands to nuclear actin polymerization thereby transducing extracellular cues for immediate nucleoskeletal consequences and genome organization.

## Methods

**Plasmids and antibodies**. Plasmids were generated and sequence-verified following standard cloning procedures. GCaMP6f was described in ref. [20]. AC-GFP and Lamin-Chromobody-GFP were from ChromoTek. nAC-mCherry and Lamin-Chromobody-mCherry were constructed by exchanging the GFP with mCherry. nAC-GFP and nAC-mCherry were ligated into pWPXL for lentiviral transduction. INF2-CAAX full length in pEGFP-C1 was from Henry N. Higgs. BFP-INF2-CAAX was cloned by exchanging the EGFP in pEGFP-C1 vector with TagBFP2. Calmodulin and its mutant was a gift from Tim Plant. mCherry-CaMBP4 and the control plasmids were published by Monaco et al.[24]. Cytoplasmic AC-mCherry-NES was cloned by adding an NES (nuclear export signal) after mCherry. INF2-DID (aa 32–266) and INF2-DAD (aa 965–1249) were cloned into EFpLink vector. α-Gα[12] and α-Gαq were from Santa Cruz. α-INF2 was from Proteintech. α-Tubulin and α-histone H3 were from Cell Signaling Technology. α-Flag conjugated with HRP and α-myc conjugated with HRP were from Sigma. α-GFP (B2) was from Santa Cruz.

**Cell lines and reagents**. NIH3T3 and HEK293 cells were maintained in DMEM medium supplemented with 10% fetal bovine serum and antibiotics. In total, 750 nM A23187 (Sigma), 0.2 U/mL thrombin (Sigma), 1.5 μM thapsigargin (Sigma), 10 mM ATP (Thermo Fischer), or 20 μM LPA (Cayman Chemical) were used as indicated. DNA plasmids were transfected with calcium phosphate methods for HEK293 and Lipofectamine 2000 (Invitrogen) for NIH3T3 cells. siRNAs targeting mouse Gα genes, mDia1, mDia2 and INF2 were from Qiagen. The targeting sequences are

siGα[q]: 5′-CTGTGGGTTGTTGAAGATAAA-3′;
siGα[11]: 5′-CCGCATCGCCACAGTAGGCTA-3′;
siGα[12]: 5′-CAGAGTGAACACACAGCCTTA-3′;
siGα[13]: 5′-AAGAATAGGCAGTATCTTTAA-3′;
siINF2: 5′-CCCGGCCTTGATGCTACAACA-3′;
simDia1: 5′-CAGGAACAGTATAACAAACTA-3′;
simDia2: 5′-CTGGACAAATTTGCCAGTATA-3′.

To delete the INF2 gene in NIH3T3 cells by CRISPR/Cas9[28], we transfected cells with the pX330A-1 × 2 vector encoding for different guide RNAs (gRNAs) directed toward the INF2 exon 2. The empty vector was used to generate control cells. Gα[q], Gα[12], and INF2 gene deletion were validated by western blotting.

**Quantitative PCR**. Quantitative PCR was performed following the previous study[29]. RNA was extracted using TRIZol (Peqlab) following the manufacturer's instructions. The reverse transcription was performed using RevertAid Reverse Transcriptase (Thermo Fischer). The obtained cDNA was quantified using a SYBR green Master Mix (Bio-Rad Laboratories). The following primers were used for mouse Gα genes:

Gα[q] (GnaqFw: GGTCGGGCTACTCTGACGA;
GnaqRv: ACTTGTATGGGATCTTGAGCGT),
Gα[11] (Gna11Fw: CAACGCGGAGATCGAGAAACA;
Gna11Rv: GCCTGCATGGCGGTAAAGAT),
Gα[12] (Gna12Fw: CGGCTGGTCAAGATCCTGC;
Gna12Rv: GCGTCCACAAGAACCCTCG),
Gα[13] (Gna13Fw: CGGAAACGCTGGTTTGAATGC;
Gna13Rv: AGATTCTGTAAGGCGATTGGTCT).
The following primers were used for mouse mDia genes:
mDia1 (mDia1Fw: GGAGATGGTGTCGCAATATCTG;
mDia1Rv: CAGGTGCATATCCCGCAAG),
mDia2 (mDia2Fw: ATGGGTTACACAGACGAGAGA;
mDia2Rv: CAGCAATAATCCGAGTCCCTC).

Relative mRNA levels were calculated using the comparative ΔΔCT model normalized to the house keeping gene GAPDH.

**Fractionation**. Cells were washed, scraped carefully, and centrifuged at 4 °C. The cell pellet was resuspended in buffer P1 (HEPES 10 mM pH 7.9, EGTA 0.1 mM, DTT 1 mM, complete protease inhibitors (Roche)). Triton X-100 (final concentration 0.5%) was added, and cells were vortexed for 10 s then centrifuged at 10,000 g for 10 min. The nuclear pellet was washed with buffer P1, then lysed in buffer P2 (HEPES 20 mM pH 7.9, glycerol 25%, NaCl 400 mM, EGTA 1 mM, DTT 1 mM, complete protease inhibitors) for 90 min at 4 °C followed by centrifugation at 16,000 g for 30 min[6]. Obtained extracts were controlled by immunoblotting for α-tubulin and histone H3.

**Immunofluorescence**. NIH3T3 cells were cultured on cover slips before fixation and permeablization. After blocking with 5% goat serum, cells were stained with INF2 antibody (Proteintech) and Lamin A/C antibody (Cell Signaling Technology). Cells were mounted in mounting media (Dako) and subjected to LSM 800 confocal microscope (Zeiss).

**Immunoprecipitation**. The primary sequence of INF2 was analyzed by a CaM-binding site prediction program (http://calcium.uhnres.utoronto.ca/ctdb/ctdb/sequence.html) to identify a putative CaM-binding site located within INF2-DID domain. Cells were transfected and harvested after 1 day and lysed in lysis buffer containing 20 mM Tris-HCl (pH 7.4), 150 mM NaCl, 2 mM EDTA, 0.1% Triton X-100, and complete protease inhibitors (Roche)[30]. Supernatants were collected after centrifugation and incubated with Flag/myc-conjugated agarose beads (Sigma) at 4 °C for 3 h, before washing and collection with SDS-PAGE loading buffer. Uncropped western blots are provided as source data in Source Data file.

**Live cell imaging and quantification**. Cells were seeded in glass-bottom dishes and transfected before imaging when necessary. Live cell imaging was performed at 37 °C, in a 5% CO$_2$ chamber. Images were acquired every 2–4 s with a LSM 800 confocal microscope (Zeiss), using the 63×/1.4 oil objective, or with a Yokogawa CSU-X1 spinning using a 100×/1.46 oil objective and Photometrics PRIME camera. Drugs were applied to the cells at the microscope while scanning. Time-lapse images were recorded and cells were counted for the assembly of nuclear F-actin. To quantify cells with nuclear F-actin, three samples in parallel were included in each experiment, and in total about 30 cells were recorded and counted. Experiments were repeated three or four times. Cells were scored as positive when nuclear F-actin structures could be detected, and as negative when no nuclear actin filaments were observed. The number of positive cells was divided by the total number of cells in the field of view, which was quantified as % of cells.

**FLIM**. Stable FLIM cells were generated using lentivirus that expresses PGK-H2B-GFP and PGK-H2B-mCherry, and FACS sorted to produce populations with homogenous expression. For FLIM-FRET experiments, cells were seeded into eight-well chambers (ibidi) and treated with DMSO, 750 nM A23817, or 0.4 U/ml Thrombin for 15 min, before being fixed in 4% PFA for 10 min. Lifetime measurements were taken on a Leica TCS SP8 system, using a white light laser with a repetition rate of 80 MHz and an excitation wavelength of 488 nm. H2B-GFP emission was detected over an emission range of 495–530 nm. Data were fitted using the FLIMfit software tool developed at Imperial College London.

**EM**. Cells were treated with DMSO, 750 nM A23817 or 0.4U/ml Thrombin for 15 min, before being high-pressure frozen (Leica EM AFS2) in 0.1 mm gold membrane carriers. Samples were then freeze-substituted (Leica EM AFS2) in a freeze-substitution acetone mix, containing 0.1% uranyl acetate and 1% osmium tetroxide, and brought to room temperature over a period of 18 h. Cells were then embedded in EPON, using standard protocols. In all, 70 -nm sections were cut using an EM UC6 microtome and diamond knife (Diatome) and stained with 3% aqueous uranyl acetate and lead citrate to enhance contrast and visualize heterochromatin. Images were acquired at ×2900 magnification, on a FEI Tecnai 12 transmission electron microscope (FEI), operated at 120 kV. For quantification of heterochromatin, nuclei and nucleoli were manually segmented. A Fiji machine learning plugin (Trainable Weka Segmentation) was used to define four classes within the image: heterochromatin (electron-dense structures), euchromatin (nuclear background), nucleolus, and background. A binary image was created to segment heterochromatin and several parameters were measured using a MATLAB script, including the proportion of the nucleus occupied by heterochromatin.

**Calcium image analysis and nuclear F-actin simulation**. Calcium images by GCaMP6f and nAC-mCherry were analyzed by MATLAB. Calcium elevation in the cytoplasm or nucleus was measured by quantifying the fluorescence intensity of GCaMP6f. We quantified nuclear F-actin by the heterogeneity of nAC-mCherry (spatial intensity variance). When there were no observable filaments in unstimulated cells, the nAC-mCherry probe was distributed uniformly (Poisson noise), and the heterogeneity (spatial intensity variance) was the lowest. As F-actin formed, the fluorescence intensity heterogeneity increased and reached the maximum when F-actin formation reached its peak. When the actin filaments disappeared, the heterogeneity returned to unstimulated levels. We normalized the heterogeneity by its changing range.

We performed a simulation to verify that the fluorescence spatial heterogeneity reflects the degree of nuclear F-actin formation. Briefly, we defined a cell nucleus as a 2D circle with a diameter of 10 μm for simplicity. The initial state presented a homogenous fluorescence signal, whose signal intensity variation was dominated by the Poisson noise. The growth rate of actin filaments was set to 1.5 μm/s[31]. We

assumed that the filaments appear in a stochastic manner within the nucleus and follow Poisson distribution (mean time: 20 s).

**Image analysis and statistics**. Other images were analyzed by Image J software (NIH), Zen software (Zeiss), or Metamorph software (Molecular Devices). Error bars (+s.e.m.) were from three or four independent experiments. Two-sided Student's $t$ test was performed when there were only two experimental conditions with normal distribution. One-way ANOVA was performed for experiments with more than two conditions. Stars in the figures indicate $p$-values from Student's $t$ test or one-way ANOVA: $*p \leq 0.05$; $**p \leq 0.01$; $***p \leq 0.001$; $****p \leq 0.0001$.

**Reporting summary**. Further information on research design is available in the Nature Research Reporting Summary linked to this article.

## Data availability
The data that support the findings of this work are available from the corresponding authors upon reasonable request. Source data for Figs. 1b–d; 2a–c; 3c–g; 4a–d; 5b, d, e–h; Supplementary Figs. 1 A, C, D; 3 A, B, D-F are provided in Source Data file.

## Code availability
Custom MATLAB code for Fig. 2d–e is available at https://figshare.com/articles/Wang_et_al_MATLAB_code/9948224.

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

## Acknowledgements
We thank Miguel A. Alonso and Henry N. Higgs for INF2 plasmids and antibody reagents, Carola Gregor and Nicolai Urban for the GCaMP6f plasmid and Uri Manor for helpful suggestions. We thank Haisen Ta for help with MATLAB quantifications and laboratory members for discussion. This work was funded by grants to R.G. (HFSP RGP0021/2016; DFG GR 2111/7-1 and the Wilhelm-Sander-Foundation 2013.149.2).

## Author contributions
Y.W. and R.G. conceived the idea and planned experiments. Y.W., A.S., B.Z., M.M., J.T., E.M.K., and C.S. designed and performed the experiments and analyzed the data; N.T. and O.T.F. provided reagents and help with lentiviral expression systems; Y.W. and R.G. wrote the paper.

## Competing interests
The authors declare no competing interests.
