## [Peer Review File · Nature Communications]

Reviewers' comments:

Reviewer #1 (Remarks to the Author):

Authors analyze here the response of NIH3T3 and HEK293 cells to various drugs affecting G Protein Coupled Receptor (GPCR) signaling. When they treat these cells with an agonist of GPCR signaling (A23187) they can detect transient accumulation of nuclear F-actin in a manner that resembles the response to serum. Using various Calcium signaling inhibiting drugs they show that this response is elicited by Calcium signaling. The burst in intracellular calcium seems to precede the pic in nuclear F-actin polymerization. Nuclear actin filaments appear to be nucleated in part from the inner nuclear envelope and depend on the presence of the nucleator Formin INF2, which accumulates in this region of the nucleus. Authors argue that this transient nuclear accumulation of F-actin promotes chromatin dynamics. This is where the work is the least convincing.

Major comments:

1/ the work lacks physiological relevance. It would be much more convincing to address these questions in a physiological context in vivo in response to an hormone or any physiological input known to activate GPCR signaling. For this reason, I would argue that the work be appropriate to a more specialist audience, such as The Journal of Cell Science for example.

2/ It is difficult to understand how nuclear F-actin was quantified in most of the figures. In the Methods authors write "the nuclear actin polymerization was measured by calculating the maximal spatial signal heterogeneity of nAC-mCherry", then how is this translated into a "% of cells" (Fig 1B, C, D; Fig 2A, B; Fig 3C,D,E,F,G) is not clear. This quantification being a major basis of the paper has to be made very clear by the authors.

3/ Related to the previous point, the percentage of cells with nuclear F-actin is variable from one experiment to the other (ranging from 50 to 80% depending on the Figure). Could the authors comment on this? Is it due to the methodology of transient cell transfection? Is it related to the levels of nAC-mCherry expression inside each nucleus or is it related to the cell-cycle stage of the cell? This is never addressed, yet the same lab has published that nuclear F-actin levels change during the cell cycle (Baarlink NCB 2017). Also I could not find in the Methods whether the cells were synchronized and at which stage in the cell cycle are the observations made. This is a key issue since their responses to GPCR activation might be cell cycle modulated.

4/ Authors argue that Calcium elevation in response to GPCR signaling precedes nuclear F-actin accumulation. Here again, how was nuclear F-actin quantified? I find a bit tricky to compare (Fig 2E) an increase in fluorescence intensity of a probe reporting for Ca-levels to an increase in F-actin detected by a probe that needs to be incorporated into polymerizing/depolymerizing filaments. The 25s delay in the two pics (Fig 2E) might just reflect the differences in probe sensitivity and ways to measure this and not the real delay in events.

5/ The involvement of the nucleator INF2 in transient nuclear F-actin polymerisation would be more convincing if authors were to present raw data of the response of the Crispr-Cas INF2 edited cells (as well as the rescued cells) to the various drug treatments (Fig 3) and not just their quantification (Fig 3C), which is problematic (see point 2).

6/ The last part of the manuscript (Fig 4) where authors try to link nuclear F-actin to chromatin dynamics is the least convincing one for the following reasons:

- the chromatin changes are subtle (Fig 4A, C) and since we can also observe cytoplasmic F-actin polymerization in response to GPCR activation (Video S1 but never quantified in the manuscript), it is a bit tricky to exclusively state that the nuclear F-actin pool is solely responsible for chromatin changes. Polymerisation of cytoplasmic filaments in response to the GPCR agonist could also impact nuclear chromatin organization.

- at which cell-cycle stage are the chromatin changes observed? This is extremely important to establish. The comparison of these changes should be done on cells in the same phase of their cell cycle.

- nuclear F-actin polymerization in response to GPCR signaling is transient and lasts only 100s (Fig 1A), yet authors are measuring chromatin changes 15 min after addition of the drugs. Why such a difference in duration of the experiments? The link between the chromatin changes and the transient burst in nuclear F-actin could be unrelated. What happens to a cell treated for 15 min with a GPCR agonist? Does it experience multiple bursts in nuclear F-actin polymerisation or only one burst?

- How is the % of heterochromatin measured in Fig 4E and 4F? raw images should be presented.

7/ It remains elusive how transient polymerization of F-actin inside the nucleus promotes the disappearance of what the authors qualify, but do not show, as electron dense structures of chromatin.

Minor comments:

- Authors should explain what is A23187 the first time they refer to it in the text since not everyone is familiar with GPCR agonists.

Reviewer #2 (Remarks to the Author):

The authors explored an important issue of the triggering of actin polymerization in the nucleus. Their data suggest that it is triggered within seconds by the release of calcium from the ER. The experiments implicated Gq/11-coupled GPCRs, Gq/11, and calmodulin/INF2 as downstream players. The authors showed that calcium ionophore A23187 can trigger polymerization of nuclear actin bypassing GPCR/Gq/11 step (as could be expected, if calcium transient are the cause), but this polymerization still requires INF2.

Overall, the data are interesting and important for cell biology. While the connection between nuclear actin and chromatin reorganization is not addressed, one cannot demand so much from a single paper. The findings open new venues of investigation and might even help develop novel small molecule drugs. Several aspects of the manuscript need to be improved.

1. Statistics: where the authors have more than two groups (virtually everywhere), Student's t-test is not valid, ANOVA or equivalent with correction for multiple comparisons should be used. While it looks like the differences the authors rely on are likely to remain valid, proper stats must be used.
2. Fig. 2B. Show the blots for the effects of siRNA knockdowns in addition to quantification.
3. Fig. S3B. What are green dots in INF2 KO cells? There supposed to be no INF2, so this might be the level of non-specific staining. As it is non-zero, the staining must be quantified and statistically analyzed.
4. Minor editing is needed. E.g., lines 159-160, "data indicates" should be "data indicate" (data is plural, datum is singular); etc.

Reviewer #1

Major comments:

1/ the work lacks physiological relevance. It would be much more convincing to address these questions in a physiological context in vivo in response to an hormone or any physiological input known to activate GPCR signaling. For this reason, I would argue that the work be appropriate to a more specialist audience, such as *The Journal of Cell Science* for example.

We thank the reviewer for his suggestion but we kindly disagree here on the significance of our study. For the first time we can show that physiological ligands that activate GPCR signaling such as LPA or Thrombin promote nuclear actin assembly. It should be noted that it is technically quite challenging to image endogenous, physiological nuclear actin filaments within intact living cells as done here. Although actin is one of the most abundant proteins in the cytoplasm it is much less prominent in the cell nucleus and hence difficult to visualize (Melak et al., 2017; Plessner and Grosse, 2019; Virtanen and Vartiainen, 2017). Current microscopy, fixation and labeling techniques as well as the highly dynamic nature of GPCR/calcium-triggered actin assembly limit the possibility of visualizing nuclear F-actin in either tissue samples from animals or in living animals. We do agree that this is a highly desirable goal and certainly one of the major challenges in the field of nuclear actin dynamics in the future.

Nevertheless, our data here are the first evidence for a regulation of nuclear actin by physiological ligands and receptors! This may have wide implications for GPCR signaling as well as in understanding GPCR agonist-evoked effects in particular as roughly half of all pharmacological drugs used in the clinics act via the GPCR signaling system.

2/ It is difficult to understand how nuclear F-actin was quantified in most of the figures. In the Methods authors write “the nuclear actin polymerization was measured by calculating the maximal spatial signal heterogeneity of nAC-mCherry”, then how is this translated into a “% of cells” (Fig 1B, C, D; Fig 2A, B; Fig 3C,D,E,F,G) is not clear. This quantification being a major basis of the paper has to be made very clear by the authors.

This appears to be a misunderstanding. The reviewer refers to the quantification method for Fig 2D and E only, which is not plotted as % of cells. Although we wrote in the Methods that “Calcium images by GCaMP6f and nAC-mCherry were analyzed by MATLAB”, we agree that maybe this has not been made clear enough in the previous and rather short manuscript version. We apologize for this and have now added a detailed section in Methods, Simulation, Image Analysis and Statistics (Page 9), describing the quantification in detail.

In the previous version of the manuscript it was stated at the beginning in the figure legend 1B that NIH3T3 cells stably expressing nAC-GFP were stimulated under the confocal microscope and that from 3 samples in parallel about 30 cells were recorded and counted for each independent experiment. Experiments were repeated 3 or 4 times. Cells were scored as positive when nuclear F-actin could be detected and as negative when no nuclear actin filaments were observed. The number of positive cells was divided by the total

number of cells in the field of view, which is quantified as % of cells. We now further added this quantification method in Methods, Live Cell Imaging and Quantification, on Page 8 of the manuscript.

We would like to state again that the quantification method for experiments in Figure 2D and E is different from the quantification method based on cell counting. For 2D and E we used a MATLAB algorithm to quantify the time sequence of the appearance for both the calcium sensor and the nuclear actin probe revealing that calcium increases occur before nuclear F-actin detection, which is supported by the data in Figure 1C using BAPTA-AM to inhibit calcium. This is not translatable into “% of cells” quantified and shown in other figures using a different quantification approach.

3/ Related to the previous point, the percentage of cells with nuclear F-actin is variable from one experiment to the other (ranging from 50 to 80% depending on the Figure). Could the authors comment on this? Is it due to the methodology of transient cell transfection? Is it related to the levels of nAC-mCherry expression inside each nucleus or is it related to the cell-cycle stage of the cell? This is never addressed, yet the same lab has published that nuclear F-actin levels change during the cell cycle (Baarlink NCB 2017). Also I could not find in the Methods whether the cells were synchronized and at which stage in the cell cycle are the observations made. This is a key issue since their responses to GPCR activation might be cell cycle modulated.

We used stably nAC-GFP or -mCherry expressing cell lines to monitor nuclear F-actin formation. Variability could be due to the different fluorescent proteins (TagGFP versus mCherry with better detection of GFP) monitored. Furthermore, additional transfections such as of siRNAs or plasmid DNAs could slightly affect the absolute number of responsive “% of cells”. We also used different stimuli to provoke nuclear actin assembly. The calcium ionophore A23187 for example constantly delivered stronger responses as may be expected in comparison to the addition of the physiological GPCR ligands used in this study.

In our imaging experiments in this study, we observed very few cells that were undergoing mitosis. To confirm that rapid stimuli-evoked effects were not related to a particular cell cycle stage, we performed DNA staining and FACS analysis of the NIH3T3 cell lines that were used in the manuscript, under exactly the same conditions as if the cells underwent the same experimental procedures as before. These data shows that less than 5% of the cells were dividing at the point of imaging or assessing nuclear F-actin (Reviewer Fig. 1).

Nevertheless, we cannot entirely rule out the possibility that some of the observed variability may be due to cell cycle dependency. To maintain cells under more physiological conditions, they were not synchronised or starved. It is important to note that the here observed nuclear actin structures are different from the ones we reported at mitotic exit or during cell spreading (Baarlink et al., 2017; Plessner et al., 2015; reviewed in Plessner and Grosse 2018). GPCR/calcium-induced nuclear F-actin is very rapid, short lived (between 60-120 seconds), and INF2-dependent. The cell cycle related nuclear F-actin structures are specific for mitotic exit. They are bundled, longer lasting for more than an hour and not

INF2-dependent (Baarlink et al., 2017, Supplementary Table 1). We apologize that this has not been clearer and have now included a sentence on Page 3 addressing that these two nuclear actin events are considered to be different as well as differently regulated. It is indeed becoming more and more evident that several different signaling inputs trigger nuclear actin events for different functions including DNA damage and repair mechanisms involving different nuclear actin assembly factors (Baarlink et al., 2013; Belin et al., 2015; Caridi et al., 2018; Schrank et al., 2018).

Reviewer Figure 1. NIH3T3 cells stably expressing nAC-GFP were seeded 48 hours before the FACS analysis. Before the measurement, cells were treated with the same drugs (e.g. A23187 or thrombin) as in the experiments. Cells were stained by NUCLEAR-ID Red DNA Stain (Enzo lifesciences). The graph shows a representative example of the red fluorescence histogram. The data were analyzed by the Dean-Jett-Fox model and the Watson model. In several measurements the G2/M population was too small to obtain a reliable analysis (both models have to identify the G2/M peak). In the other measurements the G2/M population was less than 5%.

4/Authors argue that Calcium elevation in response to GPCR signaling precedes nuclear F-actin accumulation. Here again, how was nuclear F-actin quantified? I find a bit tricky to compare (Fig 2E) an increase in fluorescence intensity of a probe reporting for Ca-levels to an increase in F-actin detected by a probe that needs to be incorporated into polymerizing/depolymerizing filaments. The 25s delay in the two pics (Fig 2E) might just reflect the differences in probe sensitivity and ways to measure this and not the real delay in events.

There is possibly a misunderstanding here. The probe is not incorporated into actin filaments. The nAC probe is a genetically encoded anti-actin nanobody, derived from the actin-chromobody from ChromoTek (Melak et al., 2017), fused to a nuclear localization sequence (Plessner et al., 2015). This tool has meanwhile been successfully used to reliably monitor endogenous nuclear F-actin (Caridi et al., 2018; Schrank et al., 2018). We apologize that this was not clear enough and we have now added a short sentence on Page 3 briefly describing the nAC again.

The probe used for calcium measurements is the widely used calcium sensor GCaMP6. GCaMP proteins consist of circularly permuted green fluorescent protein (cpGFP), the calcium-binding protein calmodulin (CaM) and CaM-interacting M13 peptide (Chen et al., 2013). Calcium-dependent conformational changes in CaM-M13 lead to increased brightness of the probe. GCaMP6f is a newer and faster variant of the GCaMP calcium

sensors (Chen et al., 2013). The sensitivity of the GCaMP probe depends on the binding sensitivity of calcium and CaM. The four Ca^{2+} binding sites of CaM are organized into two pairs; two low affinity sites ($K_d \sim 10\text{-}12 \mu\text{M}$) and two high affinity sites ($K_d \sim 1\text{-}2 \mu\text{M}$) (Hoffman et al., 2014).

Therefore, comparing the affinity of the two probes, actin-chromobody (actin binding K_d in nM range, ChromoTek, <https://www.chromotek.com/technology/alpaca-antibody-advantage/>) and GCaMP6f (calcium binding K_d in μM range), one can conclude that the sensitivity of actin probe to actin is much higher than the sensitivity of the calcium probe to calcium. The signal of the less sensitive probe (GCaMP6f) increased before the more sensitive probe (actin chromobody). This clearly indicated that the calcium signal increases before nuclear actin assembles. 25 seconds would also be a quite long period for fluorescence probe sensitivity differences.

To quantify the time sequence of calcium increase and actin polymerization in the nucleus after drug treatments in images like in Fig 2D, we could measure the increase of the fluorescent signal of calcium probe, but we could not simply measure the increase/decrease of the fluorescent signal of the actin probe because the change is not simply an increase or decrease of the signal. The change is that the homogeneously distributed fluorescent signal from the nAC-mCherry probe starts to exhibit more heterogeneous patterns that are in fact actin filaments. We could not simply count the filament numbers because the images are 2D plus time (x, y, t) not 3D plus time (x, y, z, t), due to the limitation of the experimental conditions and the speed of the imaging microscope. To reflect the change of actin polymerization within the nucleus, we therefore decided to measure the heterogeneity of the fluorescent signal from the nAC-mCherry probe. In order to prove that the method is valid, we also performed a simulation and the data are now included in the manuscript. (Suppl Video 6 and Methods section, Simulation, Image Analysis and Statistics on Page 9)

5/ The involvement of the nucleator INF2 in transient nuclear F-actin polymerisation would be more convincing if authors were to present raw data of the response of the Crispr-Cas INF2 edited cells (as well as the rescued cells) to the various drug treatments (Fig 3) and not just their quantification (Fig 3C), which is problematic (see point 2).

We agree and have now incorporated as requested one representative video of each experimental conditions from the raw data as Supplementary Videos 8-20. All the raw data related to these figures are imaging sequences over time. In total, the data size is about 20GB, in original Zeiss confocal format (.lsm or .czi). If they are to be processed into videos so that they are easily readable, the size in total will be over 100GB. Due to the file size limitations we are happy to provide all the raw data by other means, e.g. in a CD or hard drive if the journal requests us to do so.

6/ The last part of the manuscript (Fig 4) where authors try to link nuclear F-actin to chromatin dynamics is the least convincing one for the following reasons:

• the chromatin changes are subtle (Fig 4A, C) and since we can also observe cytoplasmic F-actin polymerization in response to GPCR activation (Video S1 but never quantified in the manuscript), it is a bit tricky to exclusively state that the nuclear F-actin pool is solely responsible for chromatin changes. Polymerisation of cytoplasmic filaments in response to the GPCR agonist could also impact nuclear chromatin organization.

We agree that the changes in fluorescent lifetime, as a readout of chromatin compaction, are more subtle. However, the changes are significant, highly reproducible as well as comparable to previous reports (Sherrard et al., 2018). Although we agree that potential effects of cytoplasmic actin on chromatin cannot be fully excluded we can conclude that expression of a non-polymerizing nuclear actin mutant (NLS-actin^{R62D}) prevents changes in chromatin compaction in response to these drugs, clearly indicating that nuclear actin must be involved.

• at which cell-cycle stage are the chromatin changes observed? This is extremely important to establish. The comparison of these changes should be done on cells in the same phase of their cell cycle.

Please see above as well as Reviewer Fig. 1.

• nuclear F-actin polymerization in response to GPCR signaling is transient and lasts only 100s (Fig 1A), yet authors are measuring chromatin changes 15 min after addition of the drugs. Why such a difference in duration of the experiments? The link between the chromatin changes and the transient burst in nuclear F-actin could be unrelated. What happens to a cell treated for 15 min with a GPCR agonist? Does it experience multiple bursts in nuclear F-actin polymerisation or only one burst?

We disagree that the link between chromatin changes and the transient burst in nuclear F-actin could be unrelated since inhibition of nuclear F-actin assembly, by using a non-polymerizing nuclear actin mutant (NLS-actin^{R62D}) or by inhibiting the formin actin nucleator INF2, prevented stimuli-induced changes in chromatin compaction. This strongly argues that the observed changes are nuclear F-actin dependent and a direct consequence of our cell stimulations/treatments.

• How is the % of heterochromatin measured in Fig 4E and 4F? raw images should be presented.

We apologize for not having made this clear enough. This method of quantification has been used previously (Baarlink et al., 2017; Sherrard et al., 2018), and thus was not described in detail here. This has now been corrected (Page 9 Methods, EM). Briefly, nuclei and nucleoli were manually segmented in Image J/Fiji. This image was then classified into four classes; background, nucleoli, heterochromatin (electron dense structures) and euchromatin (nuclear background). A binary image was created (black is heterochromatin, white is everything else), and several parameters were measured using a MATLAB script;

this included the proportion of the nucleus occupied by what was defined as heterochromatin. We also include representative raw images from EM in Suppl Fig. S4G.

7/ It remains elusive how transient polymerization of F-actin inside the nucleus promotes the disappearance of what the authors qualify, but do not show, as electron dense structures of chromatin.

We now provide corresponding image examples in Figure S4G.

Minor comments:

- Authors should explain what is A23187 the first time they refer to it in the text since not everyone is familiar with GPCR agonists.

A23187 is a commonly used calcium ionophore, not a GPCR agonist. We apologize for the misunderstanding and have now added a sentence on Page 3 to briefly introduce A23187.

Reviewer #2:

The authors explored an important issue of the triggering of actin polymerization in the nucleus. Their data suggest that it is triggered within seconds by the release of calcium from the ER. The experiments implicated Gq/11-coupled GPCRs, Gq/11, and calmodulin/INF2 as downstream players. The authors showed that calcium ionophore A23187 can trigger polymerization of nuclear actin bypassing GPCR/Gq/11 step (as could be expected, if calcium transient are the cause), but this polymerization still requires INF2.

Overall, the data are interesting and important for cell biology. While the connection between nuclear actin and chromatin reorganization is not addressed, one cannot demand so much from a single paper. The findings open new venues of investigation and might even help develop novel small molecule drugs. Several aspects of the manuscript need to be improved.

1. Statistics: where the authors have more than two groups (virtually everywhere), Student's t-test is not valid, ANOVA or equivalent with correction for multiple comparisons should be used. While it looks like the differences the authors rely on are likely to remain valid, proper stats must be used.

We apologize for this mistake. As suggested we have re-calculated the data using one-way ANOVA when there are more than 2 groups. We changed the significance (stars in the figures) accordingly and added text in Methods section.

2. Fig. 2B. Show the blots for the effects of siRNA knockdowns in addition to quantification.

As requested we now we show Western blots for Gq and G12 in Figure 2B. Although we tried various antibodies we failed to achieve specific signals in Western blots for G11 and G13.

Reviewer Figure 2. NIH3T3 cells stably expressing nAC-GFP were transfected with siRNAs against $G\alpha_{q/11}$ and $G\alpha_{12/13}$. After 72 hours, cells were lysed and subjected to Western blot analysis. $G\alpha_q$ and $G\alpha_{12}$ were detected using antibodies from Santa Cruz ($G\alpha_q$, SC-393 (discontinued) and $G\alpha_{12}$, SC-409). Tubulin antibody was from Cell Signaling Technologies.

3. Fig. S3B. What are green dots in INF2 KO cells? There supposed to be no INF2, so this might be the level of non-specific staining. As it is non-zero, the staining must be quantified and statistically analyzed.

We have analyzed the stainings in detail and performed statistical analysis. As reviewer 2 has expected, there appears to be unspecific background staining by the INF2 antibody but reduction in KO cells was significant (>50%) (Reviewer Fig. 3). The performance was best at the nuclear membrane. The staining was significantly reduced >70% in KO-cells compared to WT cells (Reviewer Fig. 3). This observation together with Western blots (Fig. S3C) verifies the KO of INF2.

Reviewer Figure 3. INF2 control (C) and INF2 KO (KO) cells were immunostained for endogenous INF2 and Lamin A/C. Nuclei were stained with DAPI. All images were acquired with the same microscope settings. By the Lamin and DAPI staining the cells were segmented into cytoplasm and nuclear membrane (Metamorph Software). In the respective areas the average intensities of INF2 immunostainings were measured. $n \geq 20$ cells were analyzed. Data represent mean +SEM.

4. Minor editing is needed. E.g., lines 159-160, “data indicates” should be “data indicate” (data is plural, datum is singular); etc.

We thank the reviewer for pointing this out and have made changes accordingly.

We would like to kindly mention here in addition that we replaced 2 panels of the Western blots (Figure S3A, α -Histone 3 and Figure S3D) with a different blot that we could show the original uncropped blots as Source data file and in Supplementary Figure S5.

References

Baarlink, C., Wang, H., and Grosse, R. (2013). Nuclear actin network assembly by formins regulates the SRF coactivator MAL. *Science* *340*, 864–867.

Baarlink, C., Plessner, M., Sherrard, A., Morita, K., Misu, S., Virant, D., Kleinschnitz, E.-M., Harniman, R., Alibhai, D., Baumeister, S., et al. (2017). A transient pool of nuclear F-actin at mitotic exit controls chromatin organization. *Nat. Cell Biol.* *19*, 1389–1399.

Belin, B.J., Lee, T., and Mullins, R.D. (2015). DNA damage induces nuclear actin filament assembly by Formin-2 and Spire-1/2 that promotes efficient DNA repair. *eLife* *4*.

Caridi, C.P., D'Agostino, C., Ryu, T., Zapotoczny, G., Delabaere, L., Li, X., Khodaverdian, V.Y., Amaral, N., Lin, E., Rau, A.R., et al. (2018). Nuclear F-actin and myosins drive relocalization of heterochromatic breaks. *Nature* *559*, 54–60.

Chen, T.-W., Wardill, T.J., Sun, Y., Pulver, S.R., Renninger, S.L., Baohan, A., Schreiter, E.R., Kerr, R.A., Orger, M.B., Jayaraman, V., et al. (2013). Ultrasensitive fluorescent proteins for imaging neuronal activity. *Nature* *499*, 295–300.

Hoffman, L., Chandrasekar, A., Wang, X., Putkey, J.A., and Waxham, M.N. (2014). Neurogranin alters the structure and calcium binding properties of calmodulin. *J. Biol. Chem.* *289*, 14644–14655.

Melak, M., Plessner, M., and Grosse, R. (2017). Actin visualization at a glance. *J. Cell Sci.* *130*, 525–530.

Plessner, M., and Grosse, R. (2019). Dynamizing nuclear actin filaments. *Curr. Opin. Cell Biol.* *56*, 1–6.

Plessner, M., Melak, M., Chinchilla, P., Baarlink, C., and Grosse, R. (2015). Nuclear F-actin formation and reorganization upon cell spreading. *J. Biol. Chem.* *290*, 11209–11216.

Schrank, B.R., Aparicio, T., Li, Y., Chang, W., Chait, B.T., Gundersen, G.G., Gottesman, M.E., and Gautier, J. (2018). Nuclear ARP2/3 drives DNA break clustering for homology-directed repair. *Nature* *559*, 61–66.

Sherrard, A., Bishop, P., Panagi, M., Villagomez, M.B., Alibhai, D., and Kaidi, A. (2018). Streamlined histone-based fluorescence lifetime imaging microscopy (FLIM) for studying chromatin organisation. *Biol. Open* *7*, bio031476.

Virtanen, J.A., and Vartiainen, M.K. (2017). Diverse functions for different forms of nuclear actin. *Curr. Opin. Cell Biol.* *46*, 33–38.

REVIEWERS' COMMENTS:

Reviewer #1 (Remarks to the Author):

Authors have answered all my comments hence, even though we do not share the same definition of "in vivo", I am fully satisfied by their revised work.

Reviewer #2 (Remarks to the Author):

The main interest of the manuscript is that the authors established for the first time the link between the stimulation of Gq-activating GPCRs and nuclear actin polymerization. The experiments presented are technically challenging, but the data largely support authors' conclusions. I believe that this study opens new venues in investigation of cell biology and might even have implications for therapy.

The manuscript was greatly improved in revision. The authors should be particularly commended for better explanation of the methods used. I believe that the authors adequately addressed the issues raised by both reviews.

Some editing would improve the manuscript. E.g., line 106, "succeeded" should be "followed". This can be done in proofs.

Point-by-point response to the reviewers

Reviewer #1 (Remarks to the Author):

Authors have answered all my comments hence, even though we do not share the same definition of "in vivo", I am fully satisfied by their revised work.

Reviewer #2 (Remarks to the Author):

The main interest of the manuscript is that the authors established for the first time the link between the stimulation of Gq-activating GPCRs and nuclear actin polymerization. The experiments presented are technically challenging, but the data largely support authors' conclusions. I believe that this study opens new venues in investigation of cell biology and might even have implications for therapy.

The manuscript was greatly improved in revision. The authors should be particularly commended for better explanation of the methods used. I believe that the authors adequately addressed the issues raised by both reviews.

Some editing would improve the manuscript. E.g., line 106, "succeeded" should be "followed". This can be done in proofs.

We thank both reviewers for their comments and have looked through the language of the paper and made changes accordingly.